# MULTI-INSTANCE LEARNING FOR WHOLE-SLIDE IMAGE CLASSIFICATION USING HIGHER-ORDER MOMENTS

## ABSTRACT

Whole-slide images (WSIs) contain abundant pathological information. However, the extremely high resolution and substantial redundant information in WSIs pose significant challenges for both manual analysis and artificial intelligence processing. Multi-instance learning (MIL) is currently the predominant approach, which typically focuses on aggregating low-dimensional feature representations of all patches into a single vector. If the vectors of patches are regarded as random variables, this aggregation process is essentially equivalent to estimating the first-order moment of these random vectors. However, the first-order moment alone cannot fully capture the information of the entire slide, necessitating the computation of second-order moments. Specifically, we first employ attention-based multiple instance learning (ABMIL) to calculate the attention-weighted average of patches as an estimate of the first-order moment. Concurrently, we compute the covariance matrix of the patch representation vectors across the entire slide. By aggregating the information from both the first- and second-order moments, we can greatly enhance the classification accuracy of WSIs. To improve computational efficiency, we employ DBSCAN clustering that adaptively forms large clusters for abundant normal tissues and small clusters for rare pathological regions, enabling variable-resolution processing that preserves diagnostic information while reducing computational cost. Experimental results on multiple real-world datasets demonstrate that our model significantly improves the state-of-the-art performance.

## 1 INTRODUCTION

Whole-slide images (WSIs) encapsulate a wealth of pathological information critical for medical diagnostics, such as cancer detection and classification. However, their extremely high resolution, often exceeding gigapixel scales, combined with substantial redundant information, poses significant challenges for both manual analysis by pathologists and automated processing by deep learning models (Amores, 2013; Lu et al., 2021; Song et al., 2023).

To address these challenges, multi-instance learning (MIL) has emerged as the predominant framework for WSI analysis (Ilse et al., 2018; Liu et al., 2024). In MIL, a WSI is treated as a "bag" of instances (patches), with the objective of predicting a slide-level label based on these patches. Existing MIL methods typically aggregate patch features using attention-based mechanisms (Ilse et al., 2018; Campanella et al., 2019; Li et al., 2021; Zhang et al., 2022).

From a statistical perspective, ABMIL estimates the first-order moment of patch features: $\mathbb{E}_{a_i}[\mathbf{h}_i] = \sum_{i=1}^n a_i \mathbf{h}_i$, where $a_i$ are attention weights satisfying $\sum_{i=1}^n a_i = 1$.

Despite its effectiveness, relying solely on the first-order moment limits the model's ability to capture the full complexity of WSIs. The heterogeneity of pathological patterns within a slide, including spatial and feature correlations, cannot be adequately represented by a single mean-based embedding. To overcome this limitation, we introduce higher-order statistical moments, specifically the second-order moment (i.e., the covariance matrix), to enrich the slide-level representation.

Our approach extends ABMIL by computing both first-order moments and covariance matrices to capture patch variability and inter-feature relationships. To address computational challenges,

we employ adaptive clustering via DBSCAN that groups locally similar patches with varying granularity—fine-grained clusters for rare pathological regions and coarse-grained clusters for abundant normal tissues. Both moments are computed based on cluster representations rather than individual patches. Our framework generalizes existing MIL methods, as ABMIL becomes a special case when second-order moments are omitted and each cluster contains a single patch.

The contributions of this work are threefold. First, we incorporate second-order statistical moments (covariance matrices) into MIL aggregation to better capture inter-feature relationships across patches. Second, we employ adaptive clustering that adjusts granularity based on local tissue characteristics—using fine-grained clusters for rare pathological regions and coarse-grained clusters for abundant normal tissues. Third, we validate our method on two real-world WSI datasets, demonstrating consistent improvements over strong MIL baselines.

## 2 RELATED WORK

### 2.1 MIL METHODS FOR WSI ANALYSIS

In WSI analysis, MIL methods are commonly categorized into *instance-space* and *embedding-space* approaches.

**Instance-space methods** These methods classify individual patches and then aggregate the predictions using score-level operations such as maximum selection or Top-$k$ averaging (Dietterich et al., 1997; Maron & Lozano-Pérez, 1998; Xu et al., 2019). While effective in identifying sparse and discriminative regions, they are highly sensitive to outliers and often neglect the global tissue context.

**Embedding-space methods** These methods embed each patch into a feature space and then aggregate all embeddings into a slide-level representation using operations such as mean, max, or attention-based pooling (Ilse et al., 2018). More advanced models, such as CLAM (Lu & et al., 2021) and TransMIL (Li & et al., 2021), incorporate class-aware weighting and long-range context modeling. However, the majority of these approaches rely solely on the first-order statistics, which capture only the average behavior of patch representations. As a result, they often fail to reflect feature variability and structural dependencies across patches (Schirris et al., 2022).

Moreover, most existing MIL frameworks operate at fixed patch granularity, treating all regions equally regardless of diagnostic relevance. This uniform treatment leads to inefficiency in representing rare but critical regions while overemphasizing redundant areas.

### 2.2 ADAPTIVE GRANULARITY AND CLUSTERING

The concept of adaptive granularity refers to the ability to adjust the level of detail or resolution based on local data characteristics. In WSI analysis, this means using fine-grained representation for diagnostically important regions and coarse-grained representation for less critical areas. DBSCAN (Ester et al., 1996), a density-based clustering algorithm, naturally exhibits variable cluster sizes: it forms large clusters in dense, homogeneous regions (e.g., normal tissues) and small clusters in sparse, heterogeneous areas (e.g., pathological regions). This density-adaptive property aligns well with WSI characteristics, enabling computational efficiency while preserving diagnostic information.

## 3 BACKGROUND

In MIL, WSIs are treated as *bags* containing multiple patch *instances* $\{\mathbf{x}_1, \dots, \mathbf{x}_n\}$. A neural network maps each instance to a representation $\mathbf{h}_i = f(\mathbf{x}_i)$, and attention-based MIL assigns weights to instances based on their relevance to the bag's label.

## 3.1 ABMIL: A FIRST-ORDER METHOD

ABMIL computes attention weights $a_i$ satisfying $\sum_{i=1}^{n} a_i = 1$ using (Ilse et al., 2018):

$$a_i = \frac{\exp(\mathbf{w}^\top \tanh(\mathbf{V}\mathbf{h}_i))}{\sum_{j=1}^{n} \exp(\mathbf{w}^\top \tanh(\mathbf{V}\mathbf{h}_j))},$$

where $\mathbf{w}$ and $\mathbf{V}$ are learnable parameters. This can be interpreted through the lens of first-order moments.

In ABMIL, the attention weights $a_i$ form a probability-like distribution over the instances. The bag-level representation $\boldsymbol{\mu}$ can be interpreted as the expectation of the instance representations $\mathbf{h}_i$,

$$\boldsymbol{\mu} = \sum_{i=1}^{n} a_i \mathbf{h}_i = \mathbb{E}_{a_i}[\mathbf{h}_i],$$

which is exactly the expectation, or first-order moment, of the random variable $\mathbf{h}_i$. Here, $a_i$ acts as the probability of selecting instance $i$, and $\mathbf{h}_i$ is the corresponding representation. This formulation emphasizes instances with higher attention weights, focusing on those most relevant to the bag's label. This representation is passed to a classifier to predict the bag's label.

Consider a medical imaging task, such as classifying a WSI as cancerous or not. The slide (bag) contains multiple patches (instances). The attention mechanism assigns higher weights to patches likely containing cancerous cells. The bag-level representation $\boldsymbol{\mu}$, as the expectation of patch representations, focuses on these critical patches, enabling accurate classification.

## 3.2 IMPLICATIONS AND LIMITATIONS

The expectation-based aggregation in attention-based MIL is computationally efficient and interpretable, as the weights $a_i$ highlight key instances. However, relying solely on the first-order moment (the mean) captures only the central tendency of the patch representations $\mathbf{h}_i$ and overlooks the variability and structural relationships among patches within the slide. This is a critical limitation because WSIs are highly heterogeneous, containing diverse pathological patterns that may not be fully represented by a single aggregated vector. To address this issue, the incorporation of higher-order moments, particularly the second-order moment (i.e., the covariance matrix), is proposed.

The covariance matrix of the patch representations quantifies the pairwise relationships and variability between different dimensions of the patch feature vectors across the slide. Specifically, for a set of patch representation vectors $\{\mathbf{h}_i\}_{i=1}^{n}$, where each $\mathbf{h}_i \in \mathbb{R}^d$ represents the low-dimensional embedding of a patch, the covariance matrix $\boldsymbol{\Sigma} \in \mathbb{R}^{d \times d}$ can be computed as

$$\boldsymbol{\Sigma} = \sum_{i=1}^{n} (\mathbf{h}_i - \boldsymbol{\mu})(\mathbf{h}_i - \boldsymbol{\mu})^\top,$$

where $\boldsymbol{\mu} = \sum_{i=1}^{n} a_i \mathbf{h}_i$ is the mean of the patch feature vectors (the first-order moment estimated by ABMIL). The covariance matrix $\boldsymbol{\Sigma}$ captures the second-order statistics, encoding how features covary across patches. For example, it reveals whether certain feature dimensions tend to increase or decrease together, providing insight into the underlying structure and diversity of the patch representations.

By incorporating the covariance matrix, the model gains a richer representation of the slide, as it accounts for both the average behavior (via the first-order moment) and the variability or spread of the patch features (via the second-order moment). This is particularly valuable in WSIs, where pathological patterns may exhibit complex spatial and feature correlations that are critical for accurate classification. For example, in cancer diagnosis, certain tissue regions may show correlated morphological features that are indicative of disease, which mean-based aggregation may fail to capture.

## 4 METHOD

In this section, we present our Higher-Order Multi-Instance Learning (HOMIL) framework for WSI classification. The framework addresses class imbalance and computational inefficiency by inte-

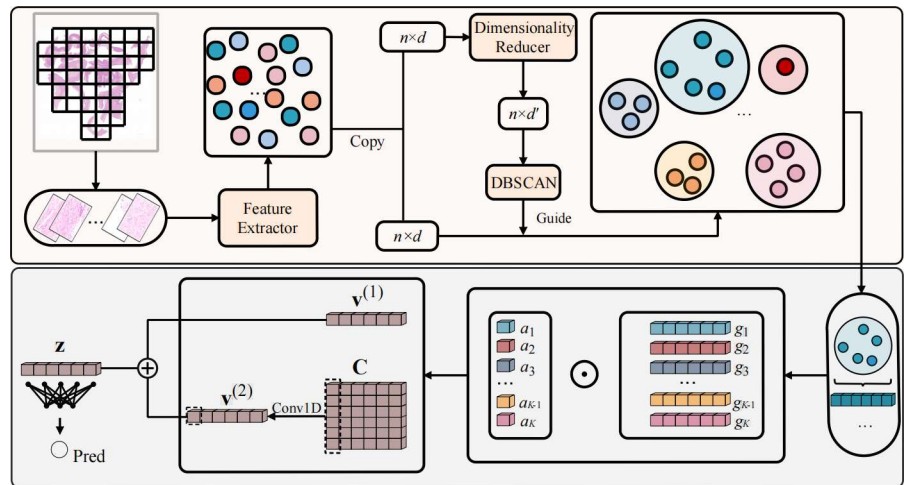

Figure 1: Overall architecture of the HOMIL framework.

grating adaptive clustering for instance aggregation and attention-weighted second-order statistics for capturing complex feature relationships.

## 4.1 OVERALL ARCHITECTURE

The HOMIL framework consists of four core modules: patch feature extraction, adaptive patch clustering, higher-order instance aggregation, and WSI-level classification. Figure 1 illustrates the overall workflow:

1. **Patch Feature Extraction**: Each WSI is first divided into non-overlapping patches of size $256 \times 256$ pixels. High-dimensional features from each patch are extracted using CONCH (Lu et al., 2024), a model pre-trained on large-scale histopathological data to capture fine-grained pathological patterns, resulting in a set of instance features $\{\mathbf{h}_i \in \mathbb{R}^d\}_{i=1}^n$ where $d = 512$.

2. **Adaptive Patch Clustering**:

- *Dimension Reduction*: Patch features are reduced from dimension $d$ to $d'$ using PCA $f_{\text{proj}} : \mathbb{R}^d \to \mathbb{R}^{d'}$, yielding $\{\hat{\mathbf{h}}_i \in \mathbb{R}^{d'}\}_{i=1}^n$.

- *Adaptive Clustering*: DBSCAN clusters the low-dimensional patches into $K$ clusters $\mathcal{C} = \{C_1, \ldots, C_K\}$, where each cluster $C_k$ contains member patches $\mathcal{I}_k \subseteq \{1, \ldots, n\}$. DBSCAN adaptively adjusts granularity: small clusters for rare pathological regions and large clusters for abundant normal tissues.

- *Cluster Feature Aggregation*: For each cluster $C_k$, patch features $\{\mathbf{h}_i | i \in \mathcal{I}_k\}$ are aggregated via mean pooling to form cluster feature $\mathbf{g}_k = \frac{1}{|\mathcal{I}_k|} \sum_{i \in \mathcal{I}_k} \mathbf{h}_i \in \mathbb{R}^d$, reducing representation from $n \times d$ to $K \times d$ where $K \ll n$.

3. **Higher-Order Instance Aggregation**: The cluster features $\{\mathbf{g}_k\}_{k=1}^K$ are processed through two parallel streams:

- *First-Order Aggregation*: Uses ABMIL to compute a weighted sum of cluster features, capturing global trends.

- *Second-Order Aggregation*: Computes an attention-weighted covariance matrix of cluster features, encoding pairwise feature correlations.

4. **WSI Classification**: The first- and second-order representations are fused via attention, and the combined vector is fed into a classifier to predict the WSI label.

## 4.2 ADAPTIVE PATCH CLUSTERING WITH DBSCAN

We employ DBSCAN (Ester et al., 1996) for adaptive patch clustering, leveraging its density-based approach that naturally aligns with WSI characteristics. Normal tissue patches, which are abundant and exhibit high feature similarity, are grouped into large clusters for coarse-grained processing. In contrast, pathological patches—rare but diagnostically crucial—form small clusters or remain as outliers, enabling fine-grained processing of critical regions.

DBSCAN is parameterized by: 1) $\epsilon$: neighborhood radius defining local density, and 2) minPts: minimum points for a dense core. Given low-dimensional patch features $\{\hat{\mathbf{h}}_i \in \mathbb{R}^{d'}\}_{i=1}^n$, clustering proceeds as:

1. For each unvisited $\hat{\mathbf{h}}_i$, compute its $\epsilon$-neighborhood $\mathcal{N}_\epsilon(\hat{\mathbf{h}}_i)$.

2. Classify $\hat{\mathbf{h}}_i$ as core if $|\mathcal{N}_\epsilon(\hat{\mathbf{h}}_i)| \geq$ minPts; otherwise non-core.

3. Form clusters by expanding from core points; non-core points form single-element clusters.

4. Each cluster $C_k$ has center $\mathbf{c}_k$, member indices $\mathcal{I}_k$, and aggregated feature $\mathbf{g}_k =$ mean$(\{\mathbf{h}_i \in C_k\})$.

## 4.3 FIRST- AND SECOND-ORDER INSTANCE AGGREGATION

Given cluster features $\{\mathbf{g}_k \in \mathbb{R}^d\}_{k=1}^K$, we compute attention weights to prioritize informative clusters, then derive first- and second-order representations for feature aggregation.

### 4.3.1 ATTENTION WEIGHT CALCULATION

For each cluster $C_k$, an attention score $a_k$ is computed to measure its importance,

$$a_k = \text{softmax}\left(\mathbf{w}_a^\top \tanh\left(\mathbf{W}_a\mathbf{g}_k + \mathbf{b}_a\right)\right) \in [0,1],$$

where $\mathbf{W}_a \in \mathbb{R}^{d_a \times d}$, $\mathbf{w}_a \in \mathbb{R}^{d_a}$, and $\mathbf{b}_a \in \mathbb{R}^{d_a}$ are learnable parameters. The weights satisfy $\sum_{k=1}^K a_k = 1$.

### 4.3.2 FIRST-ORDER REPRESENTATION

The first-order WSI embedding $\mathbf{v}^{(1)} \in \mathbb{R}^{512}$ is an attention-weighted sum of cluster features,

$$\mathbf{v}^{(1)} = \sum_{k=1}^K a_k \cdot \mathbf{g}_k,$$

which captures global trends by emphasizing clusters with high attention scores (e.g., pathological regions).

### 4.3.3 SECOND-ORDER REPRESENTATION

The second-order representation is derived from an attention-weighted covariance matrix, encoding feature correlations:

1. *Centered Features*: For each cluster $C_k$, compute the centered feature $\tilde{\mathbf{g}}_k = \mathbf{g}_k - \mathbf{v}^{(1)}$.

2. *Weighted Covariance Matrix*: The covariance matrix $\mathbf{C} \in \mathbb{R}^{d \times d}$ is computed as:

$$\mathbf{C} = \sum_{k=1}^K \tilde{\mathbf{g}}_k \tilde{\mathbf{g}}_k^\top,$$

where $\mathbf{C}$ captures how features covary across important clusters.

3. *Covariance Vectorization*: To align dimensions with the first-order representation, $\mathbf{C}$ is compressed into a $d$-dimensional vector $\mathbf{v}^{(2)}$ using row-wise 1-D convolution with multiple kernels. Specifically, for each row $\mathbf{C}_i$ of the covariance matrix $\mathbf{C}$, we convolve it with a set of $m$-dimensional kernels $\{\mathbf{k}_t\}_{t=1}^T$, where each kernel produces $d - m + 1$ outputs per row.

For each kernel $\mathbf{k}_t$, we first apply max-pooling on its output sequence to obtain a scalar $\mathbf{s}_{i,t}$,

$$\mathbf{s}_{i,t} = \max_{l=1,\dots,d-m+1} \left( \sum_{j=1}^{m} k_{t,j} \cdot \mathbf{C}_{i,l+j-1} \right).$$

Then, we perform a second max-pooling across all scalars from different kernels, resulting in a single representative scalar $\mathbf{v}_i^{(2)}$ for row $\mathbf{C}_i$,

$$\mathbf{v}_i^{(2)} = \max_{t=1,\dots,T} \mathbf{s}_{i,t}.$$

Collecting these scalars for all rows forms the final $d$-dimensional vector $\mathbf{v}^{(2)}$.

### 4.3.4 THE 1ST- AND 2ND-ORDER MOMENTS FUSION AND CLASSIFICATION

The first- and second-order representations are fused via attention to balance their contributions. For $i = 1, 2$, fusion weights are computed directly via

$$\alpha^{(i)} = \frac{\exp\left(\mathbf{w}^\top \tanh\left(\mathbf{W}\mathbf{v}^{(i)} + \mathbf{b}\right)\right)}{\sum_{j=1}^{2} \exp\left(\mathbf{w}^\top \tanh\left(\mathbf{W}\mathbf{v}^{(j)} + \mathbf{b}\right)\right)},$$

where $\mathbf{W} \in \mathbb{R}^{d_f \times d}$, $\mathbf{w} \in \mathbb{R}^{d_f}$, and $\mathbf{b} \in \mathbb{R}^{d_f}$ are shared learnable parameters, with $\sum_{i=1}^{2} \alpha^{(i)} = 1$ and $\alpha^{(i)} \in [0,1]$.

The final WSI representation is the weighted sum,

$$\mathbf{z} = \sum_{i=1}^{2} \alpha^{(i)} \cdot \mathbf{v}^{(i)} \in \mathbb{R}^d.$$

This embedding is fed into a classifier to predict the WSI label.

$$\hat{y} = \text{softmax}\left(\mathbf{W}_c \mathbf{z} + \mathbf{b}_c\right),$$

where $\mathbf{W}_c \in \mathbb{R}^{C \times d}$ and $\mathbf{b}_c \in \mathbb{R}^C$ are learnable parameters, and $C$ is the number of classes.

## 5 EXPERIMENTS

### 5.1 DATASET DESCRIPTION

We evaluate our approach on two publicly available WSI classification datasets: **CAMELYON16** and **TCGA-NSCLC**.

**1) CAMELYON16** (Bejnordi et al., 2017) is designed for metastasis detection in sentinel lymph nodes. It contains 399 H&E-stained WSIs scanned at $40\times$, with 270 for training and 129 for testing. WSIs are downsampled to $10\times$ and divided into $256\times256$ non-overlapping patches. Background is removed via Otsu thresholding, yielding roughly 3000 patches per slide.

**2) TCGA-NSCLC** (Weinstein et al., 2013) consists of 1050 H&E WSIs, split evenly between lung adenocarcinoma (LUAD, 540 cases) and lung squamous cell carcinoma (LUSC, 510 cases). Slides are scanned at $40\times$ and downsampled to $20\times$, producing an average of 15400 patches per slide.

### 5.2 BASELINES AND EXPERIMENTAL SETUP

**Baseline Methods** We compare our approach with nine MIL-based algorithms: ABMIL (Ilse et al., 2018), CLAM-SB (Lu & et al., 2021), CLAM-MB (Lu & et al., 2021), TransMIL (Li & et al., 2021), S4MIL (Fillioux et al., 2023), MambaMIL (Yang et al., 2024), HMIL (Jin et al., 2025), as well as Max Pooling and Mean Pooling. All methods are implemented in a unified codebase to ensure a fair comparison.

**Setup**  All experiments use PyTorch 2.5.1 and CUDA 12.4 with A40 GPU and AMD EPYC 7H12 CPU. All models share consistent input specifications, using patch features with a dimension of 512. For our model, the training settings include 100 epochs, learning rate $1 \times 10^{-4}$, weight decay $1 \times 10^{-5}$ and dropout rate 0.4. For both datasets, we use a unified 5-fold cross-validation setup with patient-level partitioning, where all classifiers share the same data split to ensure reliable and comparable results.

Table 1: Classification performance on CAMELYON16 (5-fold cross-validation). ACC, AUC, and F1 are reported as $\text{mean}_{\text{SE}}$ (%); Time denotes total computational time across 5 folds (seconds). Bold values indicate the best performance.

| Method | ACC (%) ↑ | AUC (%) ↑ | F1 (%) ↑ | Time (s) ↓ |
|---|---|---|---|---|
| Mean Pooling | $71.38_{8.50}$ | $73.19_{6.21}$ | $59.35_{9.41}$ | **20** |
| Max Pooling | $79.16_{2.88}$ | $83.81_{2.64}$ | $72.22_{4.90}$ | 21 |
| ABMIL (Ilse et al., 2018) | $94.72_{2.18}$ | $98.88_{1.01}$ | $93.60_{2.70}$ | 455 |
| CLAM-SB (Lu & et al., 2021) | $95.98_{3.12}$ | $98.53_{1.32}$ | $94.92_{3.97}$ | 640 |
| CLAM-MB (Lu & et al., 2021) | $95.21_{3.05}$ | $98.67_{1.20}$ | $94.27_{3.30}$ | 1115 |
| TransMIL (Li & et al., 2021) | $95.92_{1.03}$ | $98.19_{1.13}$ | $94.93_{1.17}$ | 5175 |
| S4MIL (Fillioux et al., 2023) | $95.72_{1.90}$ | $99.02_{0.87}$ | $94.61_{2.44}$ | 965 |
| MambaMIL (Yang et al., 2024) | $96.48_{1.37}$ | $98.38_{1.50}$ | $95.65_{1.75}$ | 7200 |
| HMIL (Jin et al., 2025) | $96.19_{4.18}$ | $94.44_{1.89}$ | $95.23_{2.13}$ | 10800 |
| HOMIL (Ours) | $\mathbf{96.98}_{2.43}$ | $\mathbf{99.23}_{0.62}$ | $\mathbf{96.54}_{3.03}$ | 310 |

Table 2: Classification performance on TCGA-NSCLC (5-fold cross-validation). ACC, AUC, and F1 are reported as $\text{mean}_{\text{SE}}$ (%); Time denotes total computational time across 5 folds (seconds). Bold values indicate the best performance.

| Method | ACC (%) ↑ | AUC (%) ↑ | F1 (%) ↑ | Time (s) ↓ |
|---|---|---|---|---|
| Mean Pooling | $90.76_{1.64}$ | $96.85_{0.87}$ | $90.55_{1.67}$ | 155 |
| Max Pooling | $89.71_{2.57}$ | $96.97_{1.00}$ | $89.36_{2.57}$ | **140** |
| ABMIL (Ilse et al., 2018) | $91.05_{2.05}$ | $96.58_{0.58}$ | $90.74_{2.11}$ | 4056 |
| CLAM-SB (Lu & et al., 2021) | $89.14_{3.05}$ | $95.82_{2.00}$ | $87.71_{3.95}$ | 5210 |
| CLAM-MB (Lu & et al., 2021) | $89.81_{3.53}$ | $95.62_{1.54}$ | $88.58_{4.43}$ | 5283 |
| TransMIL (Li & et al., 2021) | $88.57_{1.17}$ | $90.76_{1.50}$ | $87.80_{1.62}$ | 48710 |
| S4MIL (Fillioux et al., 2023) | $87.52_{2.36}$ | $95.58_{1.45}$ | $86.03_{2.87}$ | 4240 |
| MambaMIL (Yang et al., 2024) | $91.71_{1.64}$ | $96.68_{0.97}$ | $92.41_{1.59}$ | 25200 |
| HMIL (Jin et al., 2025) | $92.89_{1.45}$ | $93.59_{2.48}$ | $92.83_{1.47}$ | 32400 |
| HOMIL (Ours) | $\mathbf{93.24}_{2.47}$ | $\mathbf{97.41}_{1.24}$ | $\mathbf{92.93}_{2.62}$ | 3685 |

In the DBSCAN clustering algorithm, the dimension $d'$ of the linear projector $f_{\text{proj}} : \mathbb{R}^d \to \mathbb{R}^{d'}$ is set to 32. The parameter $\epsilon$ adapts to data scale: we compute distances from each data point to its nearest neighbors and set $\epsilon$ as the 65th percentile. We use minPts $= 4$ across all experimental settings.

In the stage of extracting covariance matrix information via 1-D convolution, the experiments use kernels with dimension $m = 64$ and a total of $T = 4$ kernels.

**Evaluation Metrics**  Performance is evaluated using three primary metrics for WSI classification: Accuracy (ACC), Area Under the ROC Curve (AUC), and F1 score. Additionally, we report the total computational time in seconds (including clustering for HOMIL, or training+inference only for other methods) across all 5 folds to assess efficiency. All metric results are presented as $\text{mean}_{\text{SE}}$ over a unified 5-fold cross-validation for both datasets.

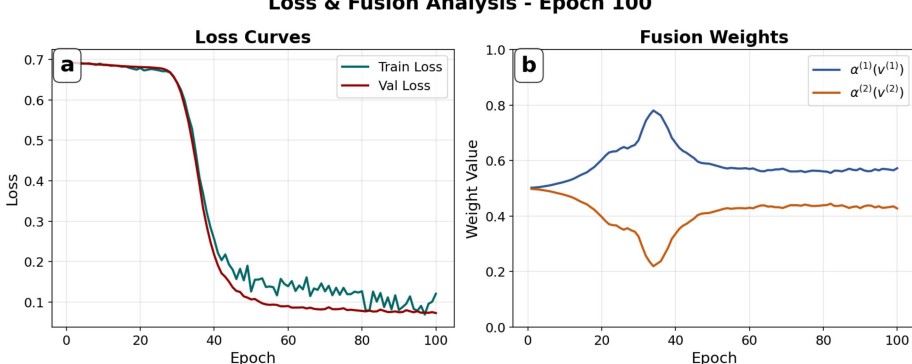

Figure 2: (a) Learning curves on the training loss and validation loss, and (b) the 1st, 2nd-order moment weights.

## 5.3 OVERALL PERFORMANCES

Under the parameter settings described earlier, the data compression ratios of the full HOMIL model on the CAMELYON16 and TCGA-NSCLC datasets are 0.18 and 0.16, respectively (calculated as the ratio of the original number of patches to the number of clusters).

The CAMELYON16 dataset, designed for lymph node metastasis detection, is a standard benchmark for WSI classification. As shown in Table 1, our method achieves the highest ACC (96.98%), AUC (99.23%), and F1 score (96.54%), outperforming all baselines. Compared to ABMIL (Ilse et al., 2018), HOMIL improves ACC by 2.26%, AUC by 0.35%, and F1 by 2.94%—findings that underscore its enhanced classification accuracy and balanced precision-recall performance. It also surpasses S4MIL in AUC (99.02%) and MambaMIL in F1 (95.65%). Notably, HOMIL exhibits superior computational efficiency with a total 5-fold runtime of 310s. This efficiency stems from adaptive clustering, which reduces complexity by grouping locally similar patches. In contrast, traditional methods like Mean Pooling and Max Pooling, though fastest (20s and 21s respectively), exhibit significantly poorer performance (ACC: 71.38% and 79.16%; F1: 59.35% and 72.22%)—a result highlighting the limitations of simplistic first-order aggregation strategies.

The TCGA-NSCLC dataset, focused on subtyping non-small cell lung cancer, presents a more complex challenge due to its histological diversity. As shown in Table 2, HOMIL achieves the best performance across all key metrics—ACC (93.24%), AUC (97.41%), and F1 (92.93%)—outperforming all baselines. Compared to ABMIL, it improves ACC by 2.19%, AUC by 0.83%, and F1 by 2.19%. Notably, in terms of computational efficiency, its total 5-fold runtime (3685s) outperforms most dynamic aggregation baselines: it is significantly faster than resource-intensive models like TransMIL (48710s), MambaMIL (25200s), and HMIL (32400s), while remaining comparable to lightweight S4MIL (4240s). This validates the efficiency of our adaptive clustering approach for large-scale WSIs.

HOMIL excels in modeling both datasets, achieving top performance across all three metrics with high efficiency. On CAMELYON16 (binary classification of metastasis) and TCGA-NSCLC (binary subtyping of lung cancer), it consistently outperforms baselines, highlighting robustness across task complexities—from metastasis detection to histological subtyping. The integration of second-order moments and adaptive clustering enhances both performance and efficiency, making HOMIL a robust and practical solution for WSI classification.

## 5.4 ABLATION STUDY

To evaluate the key components of HOMIL, we conduct an ablation study on CAMELYON16 with three variants: removing the Clustering Module (CM), disabling the Second-Order Moment module (SOM), and degenerating to ABMIL, as shown in Table 3.

The full HOMIL model, integrating CM (grouping similar patches to reduce computation) and SOM (capturing second-order statistics), achieves top performance: ACC **96.98%**, AUC **99.23%**, F1 **96.54%**, with total time 310s.

Removing CM ("w/o CM") reduces ACC by 1.26%, increases time by 71% (to 530s), and drops AUC by 1.09%, confirming its role in balancing efficiency and spatial context preservation. Disabling SOM ("w/o SOM") degrades metrics (ACC: 95.98%, AUC: 98.51%, F1: 94.94%), indicating second-order statistics capture complementary patterns. Degenerating to ABMIL (removing both CM and SOM) yields the lowest performance (ACC: 94.72%, AUC: 98.88%, F1: 93.60%) with longer time (455s), highlighting the synergy of CM and SOM.

Table 3: Ablation study on CAMELYON16 without the Clustering Module (CM) or Second-Order Moment (SOM).

| Variant | ACC (%) | AUC (%) | F1 (%) | Time (s) |
|---|---|---|---|---|
| Full model | **96.98**$_{2.43}$ | **99.23**$_{0.62}$ | **96.54**$_{3.03}$ | 310 |
| w/o CM | 95.72$_{2.61}$ | 98.14$_{2.45}$ | 94.68$_{3.29}$ | 530 |
| w/o SOM | 95.98$_{2.68}$ | 98.51$_{1.11}$ | 94.94$_{3.41}$ | **217** |
| ABMIL | 94.72$_{2.18}$ | 98.88$_{1.01}$ | 93.60$_{2.70}$ | 455 |

Results confirm both components are critical: CM enhances efficiency and spatial aggregation, while SOM enriches features via second-order statistics, together outperforming ABMIL.

## 5.5 DISCUSSION

Figure 2 (a) shows that both training and validation losses drop significantly after epoch 40 and stabilize around epoch 60, indicating efficient convergence without overfitting. In Figure 2 (b), the fusion weights for the first- and second-order moments ($\alpha^{(1)}$ and $\alpha^{(2)}$) are initially similar but diverge during training: $\alpha^{(1)}$ increases and stabilizes at a higher value, while $\alpha^{(2)}$ decreases and levels off. This suggests the model increasingly relies on first-order information, but retains second-order statistics for complementary structural cues—consistent with their joint effectiveness observed in ablation studies.

In addition, we conduct a sensitivity analysis on the key hyperparameters of the clustering module (Appendix: Sensitivity Analysis). Results show our method maintains stable, high performance across a broad range of settings, as long as the compression rate remains within a moderate range ($>$ 5%). This highlights the robustness and efficiency of adaptive clustering in balancing computational cost and diagnostic accuracy.

## 6 CONCLUSION

In this study, we propose Higher-Order Multi-Instance Learning (HOMIL), a novel framework for WSI classification that integrates first- and second-order moments with adaptive clustering. By capturing both the mean and covariance of patch features, HOMIL addresses the limitations of existing first-order methods, achieving superior performance on the CAMELYON16 and TCGA-NSCLC datasets. The incorporation of adaptive clustering enhances computational efficiency, making our approach practical for large-scale WSI analysis.

