# OpenReview forum: "Multi-Instance Learning for Whole-Slide Image Classification Using  Higher-Order Moments"
_ICLR.cc/2026/Conference — Submitted to ICLR 2026_

### Official Review · Reviewer_QK88 · 2025-10-30

**Soundness:** 2
**Presentation:** 2
**Contribution:** 2
**Rating:** 2
**Confidence:** 4

**Summary:**

This paper proposes HOMIL, a Higher-Order Multi-Instance Learning framework for WSI classification. The method incorporates adaptive clustering with DBSCAN to group locally similar patches and leverages both first-order and second-order statistical moments to aggregate information across clusters, aiming to better capture feature variability and inter-feature relationships in WSIs. The authors implement their framework against several MIL baselines on the CAMELYON16 and TCGA-NSCLC datasets, showing improvements in accuracy, AUC, and F1 score, reportedly with better computational efficiency.

**Strengths:**

1.The injection of second-order (covariance) statistics into the MIL aggregation function addresses a known limitation of first-order methods like ABMIL, potentially capturing richer inter-feature dependencies among clustered patch representations.

2.The adaptive clustering allows for significant data compression (ratios of 0.18 and 0.16 reported), helping HOMIL to achieve competitive run-times versus resource-intensive MIL models.

3.The use of DBSCAN for patch-level clustering introduces flexible granularity, allowing the model to focus on rare but critical pathological regions while reducing redundancy from normal tissues. This method provides a trade-off between computational efficiency and representation power.​

**Weaknesses:**

1.The definition of the attention-weighted covariance in Section 4.3.3 does not appear to include the attention scores $a_k$. The formula $\mathbf{C} = \sum_{k=1}^K \hat{\mathbf{g}}_k \hat{\mathbf{g}}_k^\top$ does not account for attention. Consistency with the first-order representation (which is attention-weighted) would suggest the covariance should be $\mathbf{C} \;=\; \sum_{k=1}^K a_k \, (\mathbf{g}_k - \mathbf{v}^{(1)}) (\mathbf{g}_k - \mathbf{v}^{(1)})^\top$. Without attention weights, the method loses its theoretical symmetry. This can result in large clusters being over-weighted, thereby potentially biasing the global aggregation. This should be rectified or explicitly justified.

2.There is no attention heatmap, feature embedding visualization, or per-cluster qualitative case study to help understand how the proposed model attends key WSI regions. Such visualizations are standard in recent MIL literature.

3.DBSCAN may generate many singleton or tiny clusters, especially for rare pathological regions, yet the impact of such clusters on downstream aggregation or classification is not fully analyzed. Are outliers weighted equally? Are their representations robust to class imbalance? No quantitative exploration is provided.

4.Both evaluation datasets are binary WSI tasks. The model’s scalability to multi-class or regression-based WSI settings is not explored.

5.Tables 1 and 2 report HOMIL’s performance and efficiency, but the baselines does not fully cover recent cluster-based and multi-granularity approaches. Still, the gaps over closest competitors are often small, so these tables cannot fully support claims of clear superiority.

6.Table 3 in the ablation study shows component-wise contributions and quantifies reductions in performance when clustering or second-order moments are removed, yet the total performance gains over ABMIL (the main first-order approach) is limited.

**Questions:**

1.Why is the attention weighting omitted in the calculation of the second-order covariance representation (Section 4.3.3)? Would incorporating $a_k$ as a weight for each cluster covariance improve alignment with the first-order representation and model performance? Please provide the theoretical justification or empirical comparison.

2.While the statistical motivation of using second-order moments is intuitive, a more formal analysis of why second-order moments capture critical diagnostic information is needed. For example: How do covariance eigenvalues relate to pathological heterogeneity? Could higher-order moments (e.g., skewness or kurtosis) further improve performance?

3.How does HOMIL handle singleton or very small clusters, particularly those corresponding to rare yet critical pathological regions? Are their contributions rescaled or regularized in aggregation? Is there risk of under-representation?

4.Can HOMIL be used in multi-class WSI datasets?

5.If possible, please provide attention heatmaps, cluster visualizations, or case studies showing how the HOMIL model attends to diagnostically relevant regions. This would substantiate claims of interpretability and diagnostic focus.

6.Can some alternatives to DBSCAN (e.g., hierarchical clustering) offer better control over cluster granularity?

---

### Official Review · Reviewer_jHyt · 2025-10-31

**Soundness:** 2
**Presentation:** 2
**Contribution:** 3
**Rating:** 4
**Confidence:** 4

**Summary:**

This paper identifies the limitations of the MIL paradigm and addresses them by jointly optimizing WSI classification using first-order and second-order moments. It employs clustering methods to mitigate image heterogeneity in WSI data while reducing model complexity. The proposed approach demonstrates strong performance across two publicly available datasets.

**Strengths:**

1 The model offers a new perspective on the limitations of existing MIL approaches and employs a joint optimization of first-order and second-order moments.

2 HOMIL introduces adaptive DBSCAN clustering to group patches with similar features into clusters, and implements adaptive weighted aggregation within each cluster, thereby reducing model complexity and enhancing model accuracy.

**Weaknesses:**

1 The authors provides a graphical  (Figure1) to illustrate the overall framework of the method. However, the current figure does not seem to fully align with the described approach. Specifically, what is the meaning of the “Guide” in the diagram within the method? Moreover, The details of splitting into two branches after copying should also be highlighted within the method.


2 The comparison experiments are not sufficient. Although the paper demonstrates good performance on two public datasets, this is not enough to fully illustrate the advantages of their method. More experimental comparisons are needed, such as with additional methods and datasets.


3 The current experimental analysis is insufficient and requires more visual comparative analysis to better demonstrate the advantages of their method.

**Questions:**

1 In Adaptive Path Clustering, the first step is to reduce the dimension before performing clustering. Why is the dimension of the aggregated features d rather than d'?

---

### Official Review · Reviewer_8fSw · 2025-10-31

**Soundness:** 2
**Presentation:** 2
**Contribution:** 2
**Rating:** 2
**Confidence:** 4

**Summary:**

The paper proposes using high-order moments to address the challenge that the first-order moments cannot fully capture the information from the WSIs. Specifically, this method computes the ABMIL first-order aggregation, as well as the covariance matrix of the patch representation vectors, and then aggregate these two pieces of information together to boost performance. The experimental results show the method has achieved the overall best performance against the baselines.

**Strengths:**

1.	The paper is generally easy to understand

2.	The paper presents an interesting idea that incorporating the second-order moments can enhance the model performance

3.	The experiments demonstrate the effectiveness of this method

**Weaknesses:**

1.	The paper does not contain a bibliography section.

2.	The authors should be specific about what type or kind of information is missed in first-order moments but captured in higher-order moments. In addition, why not third-order moments?

3.	The statement around line 054 to line 058 is confusing. According to line 056, both moments are calculated with cluster representations, and the following sentence says this framework is an extension to ABMIL. ABMIL calculates the first-order moments using the original patch representations, which makes these consecutive two sentences contradictory to each other.

4.	The utilization of DBSCAN requires further clarification. DBSCAN does not require an input of the number of cluster k, instead, it learns k through itself. Empirically, I have done such experiments on clustering the patch representations using DBSCAN, and the resulting ks are ranging from 5 to 30. Choosing only this number of representative tokens would cause severe information loss, unless proven otherwise. In addition, this method is not very stable, and the cluster might vary across different runs. It would be very useful if the authors could provide the clustering results and provide a more comprehensive analysis of this as well as its impact on the later stages.

5.	The method section is lengthy. The authors are advised to provide some of the essentials for the readers to understand first-order and second-order moments, not repeating ABMIL for multiple times and not using bullet points to define concepts.

6.	The experiments need improvements. There are several SOTA methods missing from comparison, such as 2DMamba, RRTMIL, CATE, DGRMIL, etc. Also, for the ablation study, at least different clustering methods with different configurations, and third-order moments should be tested.

**Questions:**

1.	Could the authors give some concrete examples for what kind of information is missed in first-order moments but captured in higher-order moments?

2.	What is the computational and memory footprint of computing covariance features, especially on slides with many clusters?

---

### Meta-Review · Area_Chair_e4pC · 2025-12-21

**Summary:**

The reviewers raised several critical concerns regarding the technical soundness, theoretical motivation, and experimental rigor of the paper. A primary issue is the overall presentation and clarity, noted by the complete absence of a bibliography and a lengthy, repetitive methodology section that lacks a clear distinction between the proposed framework and existing ABMIL approaches. The reviewers identified a potential mathematical error in the definition of attention-weighted covariance, which seems to omit attention scores; this omission undermines the theoretical symmetry of the method and risks biasing the global aggregation toward larger clusters.

The choice of DBSCAN for clustering was another major point of contention. Reviewers highlighted that DBSCAN is inherently unstable and can lead to significant information loss if the number of representative tokens is too small. There is a lack of analysis regarding how the model handles outliers or tiny clusters, and the authors did not provide enough evidence to justify why second-order moments were chosen over third-order moments or what specific information these higher-order moments capture that first-order moments do not. Additionally, the visual representations in the paper were found to be inconsistent with the text, particularly regarding the "Guide" component and the branching logic of the framework.

Finally, the experimental evaluation was deemed insufficient to support the paper's claims. The reviewers noted that the performance gains over the baseline ABMIL are marginal and that the study is limited to binary classification tasks, failing to explore multi-class or regression settings. The comparison lacks several recent state-of-the-art methods, and the ablation studies do not sufficiently test different clustering configurations. Furthermore, the absence of qualitative visualizations—such as attention heatmaps or feature embeddings—makes it difficult to verify how the model identifies and attends to key regions within the images.

**Reviewer Concerns:**

The authors did not provide a rebuttal.

**Reviewer Scores:**

The authors did not provide a rebuttal.

---

### Decision · Program_Chairs · 2026-01-26

Reject